# Common Polymorphisms Linked to Obesity and Cardiovascular Disease in Europeans and Asians are Associated with Type 2 Diabetes in Mexican Mestizos

**DOI:** 10.3390/medicina55020040

**Published:** 2019-02-05

**Authors:** Angélica Saraí Jiménez-Osorio, Claudette Musalem-Younes, Helios Cárdenas-Hernández, Jacqueline Solares-Tlapechco, Paula Costa-Urrutia, Oscar Medina-Contreras, Julio Granados, Catalina López-Saucedo, Teresa Estrada-Garcia, Martha Eunice Rodríguez-Arellano

**Affiliations:** 1Laboratorio de Medicina Genómica, Hospital Regional Lic. Adolfo López Mateos ISSSTE, Mexico City 01030, Mexico; jimenez.osorio.as@gmail.com (A.S.J.-O.); cmy10@hotmail.com (C.M.-Y.); ibi_cardenashelios@yahoo.com.mx (H.C.-H.); jsoltlapechco@gmail.com (J.S.-T.); paula.costa.urrutia@gmail.com (P.C.-U.); julgrate@yahoo.com (J.G.); 2Laboratorio de Investigación en Inmunología y Proteómica, Hospital Infantil de México Federico Gómez, Mexico City 06720, Mexico; omedina@himfg.edu.mx; 3División de Inmunogenética, Departamento de Trasplantes, Instituto Nacional de Ciencias Médicas y Nutrición Salvador Zubirán, Mexico City 14080, Mexico; 4Department of Molecular Biomedicine, CINVESTAV-IPN, Av. IPN #2508, Col. Zacatenco, Mexico City 07360, Mexico; clopezs2001@yahoo.com.mx (C.L.-S.); testrada@cinvestav.mx (T.E.-G.)

**Keywords:** type 2 diabetes, obesity, cardiovascular disease, polymorphisms

## Abstract

*Background and objectives*: Type 2 diabetes (T2D) is a major problem of public health in Mexico. We investigated the influence of five polymorphisms, previously associated with obesity and cardiovascular disease in Europeans and Asians, on T2D in Mexican Mestizos. *Materials and Methods*: A total of 1358 subjects from 30 to 85 years old were genotyped for five loci: *CXCL12* rs501120; *CDNK2A/B* rs1333049; *HNF-1*α rs2259816; *FTO* rs9939609; and *LEP* rs7799039. We used logistic regressions to test the effect of each locus on T2D in two case–control groups with obesity and without obesity. Also, linear regression models on glucose and glycated hemoglobin (HbA1c) were carried out on the whole sample, adjusted by age, gender, and body mass index. *Results*: The *CXCL12* rs501120 C allele (OR = 1.96, *p* = 0.02), the *FTO* rs9939609 A allele (OR = 2.20, *p* = 0.04) and the *LEP* rs7799039 A allele (OR = 0.6, *p* = 0.03) were significantly associated with T2D in obesity case–control group. No significant association was found in the non-obesity case–control group. The linear regression model showed that *CDNK2A/B* rs1333049 C allele (β = 0.4, *p* = 0.03) and *FTO* rs9939609 A allele (β = 0.5, *p* = 0.03), were significantly associated with HbA1c, but no association was found among the loci with the glucose levels. *Conclusions*: Polymorphisms previously linked with obesity and cardiovascular events were also associated with T2D and high levels of HbA1c. Furthermore, we must point at the fact that this is the first report where polymorphisms *CXCL12* rs501120 and *LEP* rs7799039 are associated with T2D in subjects with obesity.

## 1. Introduction

Obesity contributes to the development of hyperinsulinemia, hyperglycemia, dyslipidemia, oxidative stress, and inflammation. It is well known that subjects with obesity have an increased risk to develop type 2 diabetes (T2D) and cardiovascular events [1]. Obesity and T2D are the result of environmental and genetic factors; some families present higher incidence of that disease than others, as well as diverse Mexican Native-American groups [2]. Also, US Hispanics have higher T2D prevalence than US non-Hispanics [3]. Obesity is the main health problem in Mexico as it has the highest prevalence, worldwide, in children obesity and the second in adult obesity. Overweight and obesity affect around 70% of Mexicans (women 71.9% and men 66.7%) in the age range of 30 to 60 years old [4]. Mexico ranks fifth in T2D prevalence [5] and the search for genetic factors has taken an important place in T2D research.

Cardiovascular disease (CD) is the main cause of death among Mexican adults, and cardiovascular events are induced by obesity and/or T2D [6]. More than 20 single nucleotide polymorphisms (SNPs) have been associated with T2D [7], but polymorphisms related to cardiovascular risk have been poorly investigated in Mexican patients with T2D. Thus, we chose five SNPs associated with obesity and CD, to assess their association with T2D, namely, *CXCL12* (stromal cell-derived factor 1 precursor) rs501120, *CDKN2A/B* (kinase inhibitor 2A) rs1333049, *HNF-1*α (hepatocyte nuclear factor-1alpha) rs2259816, *FTO* (fat mass and obesity) rs9939609, and *LEP* (leptin) rs7799039. 

The *CXCL12* gene is involved in macrophage recruitment, and it is a factor required for obesity-induced adipose tissue inflammation and systemic insulin resistance [8]. The SNP rs501120 of *CXCL12* is associated with coronary artery disease [9,10], and the progression of coronary atherosclerotic plaque in T2D Chinese individuals [11]. The *CDKN2A/B* gene is expressed in pancreatic beta cells and is related to insulin-deficient diabetes [12]. The rs1333049 polymorphism is associated with CD in European and Asian subjects [9,13,14,15,16], atherosclerosis in Italians [17], myocardial infarction in Japanese [18], and T2D and metabolic syndrome in European populations [19]. *HNF-1*α regulates the expression of key genes involved in β cell glucose-sensing, while SNP rs2259816 has been associated with T2D and CD in Americans [20] and with high levels of low-density lipoprotein cholesterol in Saudi subjects [21]. *FTO* rs9939609 shows the strongest association reported with obesity in several populations and ethnic groups [22,23,24]. Finally, *LEP* rs7799039 has been associated with higher levels of serum leptin and adipose tissue leptin secretion rate [25], obesity [26], and hypertension [27].

To contribute to a better understanding of the basis of T2D genomics in Mexico, we evaluated if the selected common polymorphisms, previously linked to obesity and cardiovascular disease in European and Asian populations, are associated with T2D in a Mexican Mestizo population.

## 2. Materials and Methods

### 2.1. Study Design

This study was carried out in 1358 men and women ranging from 30 to 85 years old from the Hospital Regional Lic. Adolfo López Mateos–ISSSTE (Instituto de Seguridad y Servicios Sociales de los Trabajadores del Estado) and the Automated Detection and Diagnosis Clinic (Clínica de Detección y Diagnóstico Automatizado, CLIDDA–ISSSTE). The anthropometric measurements weight, height, and body mass index (BMI) were recorded. Glucose and glycated hemoglobin (HbA1c) levels were determined in whole blood samples (20 mL) collected from subjects after 8 h of fasting. Obesity is defined by World Health Organization (WHO) as BMI ≥30 kg/m^2^ [28]. T2D criteria for classification was fasting plasma glucose (FPG) ≥126 mg/dL and HbA1c ≥6.5% [29]. 

Two pairs of case–control groups were formed. The first one involved subjects classified with obesity and T2D (case group: BMI ≥ 30 kg/m^2^, FPG ≥ 126 mg/dL, HbA1 ≥ 6.5%), and with obesity but without T2D (control group: BMI ≥ 30 kg/m^2^, FPG ≤ 125 mg/dL, HbA1c ≤ 6.4%). The second one involved subjects without obesity and with T2D (case group: BMI < 30 kg/m^2^, FPG ≥ 126 mg/dL, HbA1 ≥ 6.5%), and with neither obesity nor T2D (control group: BMI < 30 kg/m^2^, FPG ≤ 125 mg/dL, HbA1c ≤ 6.4%). All patients with T2D were recruited in specialized clinics for diabetes treatment in the ISSSTE, since they were previously diagnosed with T2D.

Exclusion criteria were subjects with previous history of myocardial infarction, or with foreign parents and grandparents. All participants were asked to sign an informed consent. This study was approved by the Hospital Regional Lic. Adolfo López Mateos Research, Ethics, and Biosafety Committees (registration number 236.2011), and conducted in accordance with the Declaration of Helsinki.

### 2.2. DNA Extraction and Genotyping

Genomic DNA was obtained from 500 μL of whole blood–EDTA with the InviMag Blood DNA, Stratec Mini Kit (Berlin, Germany) using an automated nucleic acid sample isolation (InviGenius, Stratec; Berlin, Germany). The five selected SPNs were genotyped by using a pre-designed 5′ exonuclease TaqMan genotyping assay on a 7500 series Real-Time PCR system, according to the manufacturer’s instructions (Applied Biosystems, Foster City, CA, USA).

### 2.3. Statistical Analysis

Descriptive results are presented by median and interquartile range due to non-normal distribution. Measurement comparisons between cases and control groups were carried out using Mann–Whitney test. The Hardy–Weinberg equilibrium on genotype distribution was evaluated using X^2^ test. We used logistic regression under dominant, recessive, and additive inheritance models to evaluate the effect of the five SNPs on T2D. All analyses were adjusted by gender, age, and BMI. In addition, a linear regression model was performed to test the effect of locus on glucose and HbA1c as response variables, adjusted by age, gender, and BMI. All statistical analyses were performed using STATA12 (StataCorp, College Station, TX, USA). 

Power estimation was conducted using case–control outcome design and 12% of T2D prevalence (as implemented in Quanto Software (USC Biostats, Los Angeles, CA, USA)). The power calculation was assessed using gene only hypothesis with SNP frequencies from 0.1 to 0.5. Our sample size provides an 80% statistical power to detect a significant association (α < 0.05) considering odds ratio (OR) values from 1.2 to 1.5.

## 3. Results

Clinical and anthropometrical values for obesity and non-obesity groups are summarized in Table 1. Diabetic subjects were consistently older than subjects without T2D. In addition, both groups showed no significant BMI differences between T2D and non-T2D subjects.

The *CXCL12* rs501120 C, *FTO* rs9939609 A, and *LEP* rs7799039 A alleles were significantly associated with T2D in the obesity group under a recessive inheritance model (Table 2). The *CXCL12* rs501120 C allele and the *FTO* rs9939609 A allele were risk factors (rs501120: OR = 1.96, *p* = 0.02; rs9939609: OR = 2.2, *p* = 0.04), while *LEP* rs7799039 was a protective factor (rs7799039: OR = 0.6, *p* = 0.03) in the obesity group. No significant association was found with any locus studied in the non-obesity group (Table 3).

The linear regression model analysis showed that the *CDNK2A/B* rs1333049 C and the *FTO* rs9939609 A alleles were significantly associated with HbA1c. No significant association was found with glucose levels and the SNPs included in this study (Table 4).

## 4. Discussion

Obesity is a metabolic disorder that increases the risk of T2D and CD. As a consequence, polymorphisms linked to obesity and cardiovascular events are likely associated with T2D [6]. Our results showed that *CXCL12* rs501120, *LEP* rs7799039, and *FTO* rs9939609 polymorphisms were associated with T2D in subjects with obesity, but not in subjects without obesity. 

We found that the *CXCL12* rs501120 C allele was a risk factor for T2D in Mexican subjects with obesity and without CD. This polymorphism has been widely studied in several populations. In Caucasians, this locus was associated with coronary artery disease [9,30]. In Chinese male subjects, the C allele was associated with an increased risk of ischemic stroke [31]. Kiechl et al. reported that T carriers showed lower plasma CXCL12 levels than C carriers [32], leading to lower inflammation in non-diabetic subjects with obesity. However, Mehta et al. reported that T carriers present higher levels of *CXCL12* in plasma, in a bigger sample size [33]. Although the results in diverse populations have been controversial, it is accepted that *CXCL12* is a protein that promotes the secretion of proinflammatory cytokines and the development of insulin resistance [34,35]. Although mRNA and protein expression have not been determined, some studies suggest that other polymorphisms (rs1746048 C) may be related to increased expression of CXCL12 [33]. Therefore, our results suggest that this polymorphism could increase the proinflammatory action of *CXCL12* in subjects with obesity, increasing the risk of T2D, but molecular mechanisms need to be evaluated to confirm this suggestion.

Additionally, our results showed that the LEP rs7799039 A was a protective factor against T2D in subjects with obesity. In Caucasian individuals, AA carriers showed higher body fat and BMI [36]. In Southern Chilean individuals, the A allele was associated with a decrease in total and low-density lipoprotein cholesterol. In Brazilian pregnant women, the A carriers showed lower body weight during pregnancy [37]. In Mexican women, G carriers had higher leptin levels in serum, while A carriers showed lower risk of obesity [38], suggesting that the A allele is an obesity protective factor in Hispanic populations. *LEP* is an adipokine that regulates food intake by suppression of appetite, and high *LEP* levels in serum are associated with inflammation, oxidative stress, and the development of CD and insulin resistance [39]. In this study, we found that rs7799039 was more frequent in subjects with obesity and without T2D. The increase of fat mass leads to higher leptin levels, inducing a proinflammatory response and insulin resistance [40]. The SNP rs7799039 could impair LEP action diminishing the insulin resistance effect derived by a leptin resistant state during obesity.

The common *FTO* rs9939609 polymorphism has been strongly linked to BMI and body fat in several populations [23,41,42,43,44]. In this context, the study of this polymorphism can be considered as a positive control, due to the fact that the locus has been associated with obesity in Mexican populations [45,46], and the risk alleles showed a nominal association with lower insulin levels and the homeostatic model assessment of beta cell function (HOMA-β) [47]. Interestingly, this is the first report where the rs9939609 polymorphism is associated with T2D and HbA1c levels in Mexican subjects with obesity. However, little is known about the molecular mechanisms that *FTO* uses to induce obesity and T2D. In a recent study, *FTO* protein expression in lymphocytes was higher in patients with severe T2D, and correlated with waist and hip circumference, BMI, fasting blood glucose, and HbA1c [48]. Although the specific molecular mechanisms of how the *FTO* gene increases HbA1c or disturb glucose homeostasis are not fully understood, it can be suggested that epigenetic factors may play an important role. Therefore, it is crucial to elucidate the role of the rs9939609 polymorphism in *FTO* expression and glucose homeostasis. Thus, our results suggest that A carriers interfere with *FTO* activity in subjects with obesity, increasing the HbA1c levels in T2D subjects. 

Finally, we found a positive association between HbA1c levels and the *CDNK2A/B* C allele in adults without obesity. Previous reports show that this polymorphism is associated with fasting insulin, the homeostatic model assessment of insulin resistance (HOMA-IR), and insulin secretion in Chinese T2D subjects. However, in this study, we observed a weak association between this locus and T2D in subjects without obesity. This observation correlates with previous studies showing these polymorphisms are specific to CD disease [49].

This study presented limitations that need to be mentioned. Diagnosis of diabetes based on FPG and HbA1c is frequently sufficient, however, there are patients whose FPG is <125 mg/dL and HbA1c is ≤6.4% and who present diabetes, since their glucose level after 2 h of oral glucose tolerance test (OGTT) is >200 mg/dL [29]. As we did not perform 2 h of OGTT, such cases could not be detected, if they have occurred.

## 5. Conclusions

In this study, we showed that common polymorphisms linked to cardiovascular events and obesity in Europeans and Asians are also associated with T2D in Mexican Mestizo subjects with obesity. Furthermore, this is the first report that highlights an association between polymorphisms in *CXCL12* rs501120 and *LEP* rs7799039 with T2D in Mexican Mestizo adults with obesity.

## Figures and Tables

**Table 1 medicina-55-00040-t001:** Clinical and anthropometrical values.

	Obesity Group			Non-Obesity Group		
Variable	Case	Control	*p*-Value	Case	Control	*p*-Value
*N*	301	321		336	400	
Age	54 (47–62)	48 (45–52)	<0.001	55 (49–64)	49 (45–53)	<0.001
Male, *n* (%)	134 (44)	189 (57.5)	0.001	189 (54)	211 (50.7)	0.365
BMI	33.4 (31.3–36.4)	32.8 (31.2–35)	0.1201	26.6 (24.1–28.3)	26.4 (24.7–28.1)	0.7958
Glucose	128 (100.5–187)	98 (91.5–105)	<0.001	121.3 (99–164.8)	94 (89–101)	<0.001
HbA1c	5.8 (4.9–7.4)	4.2 (3.6–4.9)	<0.001	5.85 (4.7–7.4)	4 (3.5–4.6)	<0.001

Data are presented as median (interquartile range) for continuous variables. *N*: number of individuals per group, BMI: body mass index, HbA1c: glycated hemoglobin.

**Table 2 medicina-55-00040-t002:** Logistic regression of polymorphisms studied in obesity case–control group.

SNP	Gene	MAF	Allele	OR_DOM_ (95% CI), *p*	OR_REC_ (95% CI), *p*	OR_ADD_ (95% CI), *p*
rs501120	*CXCL12*	0.3	C	1.02 (0.7–1.4), 0.88	1.96 (1.1–3.7), 0.02	1.15 (0.8–1.5), 0.3
rs1333049	*CDNK2A/B*	0.46	C	1 (0.7–1.4), 0.99	0.8 (0.5–1.2), 0.29	0.93 (0.7–1.2), 0.54
rs2259816	*HNF-1*α	0.37	T	0.8 (0.6–1.1), 0.25	1.02 (0.6–1.6), 0.92	0.9 (0.7–1.2), 0.45
rs9939609	*FTO*	0.34	A	1.03 (0.7–1.4), 0.84	2.2 (1.1–5.1), 0.04	1.14 (0.85–1.5), 0.36
rs7799039	*LEP*	0.22	A	0.84 (0.6–1.2), 0.32	0.6 (0.3–0.9), 0.03	0.8 (0.6–1.02), 0.07

Inheritance Models = DOM: dominant, REC: recessive, ADD: additive. MAF: minor allele frequency.

**Table 3 medicina-55-00040-t003:** Logistic regression of polymorphisms studied in non-obesity case–control group.

SNP	Gene	MAF	Allele	OR_DOM_ (95% CI), *p*	OR_REC_ (95% CI), *p*	OR_ADD_ (95% CI), *p*
rs501120	*CXCL12*	0.3	C	1.05 (0.7–1.4), 0.78	0.92 (0.5–1.7), 0.81	1.01 (0.8–1.3), 0.9
rs1333049	*CDNK2A/B*	0.48	C	0.85 (0.6–1.2), 0.33	1.47 (1–2.2), 0.06	1.1 (0.8–1.3), 0.52
rs2259816	*HNF-1*α	0.4	T	1.03 (0.7–1.4), 0.41	0.82 (0.5–1.3), 0.41	0.96 (0.7–1.2), 0.75
rs9939609	*FTO*	0.41	A	0.52 (0.2–1.2), 0.13	1.9 (0.8–4.4), 0.13	1.1 (0.8–1.4), 0.51
rs7799039	*LEP*	0.19	A	0.93 (0.6–1.3), 0.69	1.27 (0.8–1.9), 0.26	1.03 (0.8–1.3), 0.74

Inheritance Models = DOM: dominant, REC: recessive, ADD: additive. MAF: minor allele frequency.

**Table 4 medicina-55-00040-t004:** Linear regression model of glucose and Hba1c as response variable including polymorphisms studied as additive model in subjects with obesity.

SNP	Gene	MAF	Allele	β Glucose (95% CI)	*p*	β HbA1c (95% CI)	*p*
rs501120	*CXCL12*	0.30	C	3.5 (−4.8–11.8)	0.41	−0.1 (−0.5–0.3)	0.59
rs1333049	*CDNK2A/B*	0.46	C	−2.6 (−9.9–4.7)	0.48	0.4 (0.04–0.7)	0.03
rs2259816	*HNF-1*α	0.37	T	−4.6 (−12.5–3.2)	0.25	0.07 (−0.3–0.4)	0.68
rs9939609	*FTO*	0.34	A	0.94 (−10.1–8.2)	0.84	0.5 (0.4–0.9)	0.03
rs7799039	*LEP*	0.22	A	−1.6 (−9.3–6)	0.67	0.3 (−0.05–0.6)	0.09

Each model was adjusted by age, gender and BMI. MAF: minor allele frequency.

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
