# Peer review of "Common Polymorphisms Linked to Obesity and Cardiovascular Disease in Europeans and Asians are Associated with Type 2 Diabetes in Mexican Mestizos"

_medicina, 2019, doi:10.3390/medicina55020040_

Round 1
Reviewer 1 Report
The work described in the current manuscript has been competently executed and thoroughly analysed. Whilst many much larger studies have produced data on these SNPs with much greater significance there is a growing consensus within the genetics community that further association analyses must also be carried out on non-Asian and European gene pools. The current study takes a small step in realising this need by exploring the associations of obesity associated SNPs in a Mexican population.
Author Response
We are very thankful for the useful and constructive comments.
The introduction was improved and english language was edited.
Reviewer 2 Report
Reviewer's report
Manuscript title:
Common polymorphisms linked to obesity and cardiovascular disease in Europeans and Asians are associated with Type 2 Diabetes in Mexican-Mestizos
Authors:
Angélica Saraí Jiménez-Osorio , Claudette Musalem-Younes , Helios Cárdenas-Hernández , Jacqueline Solares-Tlapechco , Paula Costa-Urrutia , Oscar Medina-Contreras , Julio Granados , Catalina López-Saucedo , Teresa Estrada-García , Martha Eunice Rodríguez-Arellano.
Obesity and its complications reach pandemic proportions; therefore there is a constant need for studies on the factors that could predispose toward excessive weight gain. The idea of genetic studies, like this one, is to identify individuals at high risk of obesity and its complications. In this context the idea of the presented work is reasonable. The studied group size and homogeneity, as well as genotyping methods, are also appropriate. However, I have some remarks regarding the article structure and conception.
Major Remarks
Layout and format
In general, the paper meets the Medicina guidelines regarding the manuscript presentation with two exceptions:
a) The section Materials and Methods should be moved after the Discussion
b) In the Back Matter, Acknowledgements should be placed before Author Contribution
c) The References do not meet the “Reference style” recommended by the Medicina guidelines including the fact that the reference numbers are not placed in square brackets
Title and abstract
Quite accurately describes the article and its content.
Introduction
Apart from the epidemiological data regarding the prevalence of obesity and related complications in Mexican adults the Authors should explain why the particular genes and polymorphisms were selected for the analysis. With this in mind, they can remove the whole paragraph from the Materials and Methods section (lines 83-91) and supplement it by short information about the possible pathogenic mechanisms linking the selected SNPs with obesity.
Materials and Methods
a) Study design: although BMI is commonly used for the diagnosis and staging of obesity it does not refer to body composition. To define a phenotype of obesity and the risk of related complications adipose tissue content and distribution are more appropriate. It would be of value to know if adipose tissue content and, e.g. waist circumference were measured in the participants of the study.
b) Diagnosis of diabetes based on FPG and HbA1c is frequently sufficient; however there are patients whose FPG is < 125 mg/dl and HbA1c is ≤ 6.4% and who have diabetes since their glucose level after 2 hours of OGTT is > 200 mg%. It mainly refers to non-obese and elderly subjects and those with FPG < 125 mg/dl and HbA1c is > 6.4%. Were there any subjects in the studied group? ADA Guidelines state:
“The concordance between the FPG and 2-h PG tests is imperfect, as is the concordance between A1C and either glucose-based test. Numerous studies have confirmed that compared with FPG and A1C cut points, the 2-h PG value diagnoses more people with diabetes” Diabetes Care 2016 Jan; 39(Supplement 1): S13-S22.
Therefore, using only FPG and HbA1c for the diagnosis of T2D, there is a risk that cases are not separated from the controls.
c) The Authors state that "Exclusion criteria were subjects with CD" – it would be essential to explain how the diagnosis was established? What diagnostic tests applied?
Statistical analysis
I am not an expert in the field, however, if the data distribution is non-normal – shouldn't the descriptive result be presented as medians? Iran J Public Health. 2015 Nov;44(11):1557-8.
Results
This section together with the tables attached is clear, apart from the fact that the caption for Table 4 is misplaced into Discussion
Discussion
In this section, the Authors should try to find the possible explanation for the obtained results and discuss if the investigated variants have any functional consequences. Moreover, some sentences are unclear:
Line 146 – since inflammation constitutes a link between obesity and T2D why impairment of pro-inflammatory action of CXCL12 should protect from obesity?
Lines 150-154 – what mechanism the Authors suggest to explain how LEP rs7799039 A allele increases the risk of T2D in the studied population while in other populations it was found to protect from obesity and unfavourable lipid profile?
Language
The paper may benefit from the assistance of the native speaker.
Minor Remarks
1) The names of genes should be written in italics
2) I am not a native, however:
- the singular for loci is locus – so, e.g. line 26 “the effect of each loci” should be replaced by “the effect of each locus” as well as in the lines 139, 160 and 175
- line 31 “lineal model” should rather be replaced by “linear model”
- line 43 “Obesity and T2D are the result” should rather be replaced by “Obesity and T2D are the results”
- line 45 “more T2D prevalence” should rather be replaced by "higher T2D prevalence."
Author Response
Layout and format
Q1. The section Materials and Methods should be moved after the Discussion
R1.The section “Materials and Methods” was placed in section 2 according to the medicina template (Microsoft Word Template). The last published articles had this format. Please, let us know if this is a new instruction to make the appropriate change.
Q2. In the Back Matter, Acknowledgements should be placed before Author Contribution
R2.The subsection Acknowledgements was placed before Author Contribution.
Q3.The References do not meet the “Reference style” recommended by the Medicinaguidelines including the fact that the reference numbers are not placed in square brackets
R3.All the references were reviewed and were included according to Medicina guidelines. The reference numbers in the text were placed in square brackets.
Title and abstract
Comment: Quite accurately describes the article and its content.
R: Thanks for this comment
Introduction
Q4. Apart from the epidemiological data regarding the prevalence of obesity and related complications in Mexican adults the Authors should explain why the particular genes and polymorphisms were selected for the analysis. With this in mind, they can remove the whole paragraph from the Materials and Methods section (lines 83-91) and supplement it by short information about the possible pathogenic mechanisms linking the selected SNPs with obesity.
R4:The lines 89-91 (previous format) were moved to Introduction (lines 54 to 71). Additionally, we included the information of possible pathogenic mechanisms.
Q5. Study design: although BMI is commonly used for the diagnosis and staging of obesity it does not refer to body composition. To define a phenotype of obesity and the risk of related complications adipose tissue content and distribution are more appropriate. It would be of value to know if adipose tissue content and, e.g. waist circumference were measured in the participants of the study.
R5: Thanks for the observation. The measure of body fat mass and waist circumference was not available for all the individuals. However, BMI measure is well accepted as a good criterion for obesity classification in adults (WHO expert consultation 2004).
Q6.Diagnosis of diabetes based on FPG and HbA1c is frequently sufficient; however there are patients whose FPG is < 125 mg/dl and HbA1c is ≤ 6.4% and who have diabetes since their glucose level after 2 hours of OGTT is > 200 mg%. It mainly refers to non-obese and elderly subjects and those with FPG < 125 mg/dl and HbA1c is > 6.4%. Were there any subjects in the studied group? ADA Guidelines state:
“The concordance between the FPG and 2-h PG tests is imperfect, as is the concordance between A1C and either glucose-based test. Numerous studies have confirmed that compared with FPG and A1C cut points, the 2-h PG value diagnoses more people with diabetes” Diabetes Care 2016 Jan; 39(Supplement 1): S13-S22.
Therefore, using only FPG and HbA1c for the diagnosis of T2D, there is a risk that cases are not separated from the controls.
R6.Thanks for this observation, this is an important point. We included in the section Materials and Methods the explanation that all the patients with diabetes were previously diagnosed by specialized clinics in diabetes of the ISSSTE (line 80-90). T2D patients are diagnosed in this clinics with 2h PG, however data was not available for all subjects.With this in mind, we included a statement in Discussion about the limitations of the study according to this criterion for selection.
Q7.The Authors state that "Exclusion criteria were subjects with CD" – it would be essential to explain how the diagnosis was established? What diagnostic tests applied?
R7.We excluded subjects with previously events of myocardial infarction. Changes were made in Material and Methods, and we wrote “Exclusion criteria were subjects with previous history of myocardial infarction”.
Q8. I am not an expert in the field, however, if the data distribution is non-normal – shouldn't the descriptive result be presented as medians? Iran J Public Health. 2015 Nov;44(11):1557-8.
R8.It is true. All the variables had a non-normal distribution and were expressed as medians. The results expressed in Table 1 are the medians of each continuous variable instead of means. Changes were made in Table 1 (foot of table) and statistical analysis subsection.
Results
Q9.This section together with the tables attached is clear, apart from the fact that the caption for Table 4 is misplaced into Discussion
R9.Thanks for the observation. Table 4 was placed in the Results section.
Discussion
In this section, the Authors should try to find the possible explanation for the obtained results and discuss if the investigated variants have any functional consequences. Moreover, some sentences are unclear:
Q10.Line 146 – since inflammation constitutes a link between obesity and T2D why impairment of pro-inflammatory action of CXCL12 should protect from obesity?
R10.There was a misinterpretation. The effect of the rs501120 in CXCL12 mRNA and protein expression and activity are not well understood. However, some studies suggest that other polymorphisms in this gene increase the mRNA expression of CXCL12, increasing proinflammatory action. In our study, we found that the rs501120 polymorphism increases the risk of T2D in subjects with obesity. Thus, we suggest that a proinflammatory action of CXCL12 is exacerbated.
Q11. Lines 150-154 – what mechanism the Authors suggest to explain how LEP rs7799039 A allele increases the risk of T2D in the studied population while in other populations it was found to protect from obesity and unfavourable lipid profile?
R11. Thanks for this observation, there was a mistake in the following sentence:
“Our results showed that the LEP rs7799039 A allele increased the T2D risk in subjects with obesity”.
The sentence was corrected to: “our results showed that the LEP rs7799039 A was a protective factor against T2D in subjects with obesity” according to Table 2.
In relation to the biological mechanisms, we extended the discussion and we suggested that, due to the fact that rs7799039 was more frequent in subjects with obesity and without T2D,the SNP rs7799039 could impair LEP action, diminishing the insulin resistance effect derived by a leptin resistant state during obesity.
Language
Q12.The paper may benefit from the assistance of the native speaker.
R12.The paper was reviewed and corrected by a English native speaker
Minor Remarks
Q13. The names of genes should be written in italics
R13.Thanks for the observation. Gene names were wrote in italics
Q14.I am not a native, however:
- the singular for loci is locus – so, e.g. line 26 “the effect of each loci” should be replaced by “the effect of each locus” as well as in the lines 139, 160 and 175
- line 31 “lineal model” should rather be replaced by “linear model”
- line 43 “Obesity and T2D are the result” should rather be replaced by “Obesity and T2D are the results”
- line 45 “more T2D prevalence” should rather be replaced by "higher T2D prevalence."
R14.Thanks for all suggestions, the changes were made in the extensive review of the English language.

Round 2
Reviewer 2 Report
None
Author Response
We are very thankful for the useful and constructive comments of your review. We have carefully considered all of them, and made the necessary changes in the revised manuscript.
Sincerely, on behalf of the authors,
Eunice Rodríguez-Arellano